# Effect of Fresh Dendrobe Juice Consumption on Senile Habitual Constipation for Older People: A Four-Week Randomized Controlled Trial

Yuchao Le [1], Shihua Cao [1,2,*], Mengxin Wang [1], Danni He [1], Yanfei Chen [1] and Beiying Qian [1]

[1] School of Nursing, Hangzhou Normal University, Hangzhou 310012, China; 2020112012118@stu.hznu.edu.cn (Y.L.); 2019112012057@stu.hznu.edu.cn (M.W.); 2021112012182@stu.hznu.edu.cn (D.H.); 2021112012203@stu.hznu.edu.cn (Y.C.); 2021111012057@stu.hznu.edu.cn (B.Q.)

[2] Department of Nursing, Hangzhou Normal University Qianjiang College, Hangzhou 310036, China

\* Correspondence: csh@hznu.edu.cn; Tel.: +86-0571-2886-1972

**Abstract:** Background: In Chinese medicine, it is believed that fresh dendrobe juice, which is mild in nature, can relieve the symptom of constipation. Methods: A pilot research design was conducted. Fifty-six older people with senile habitual constipation were recruited and randomly enrolled into control or experimental groups, each with 28 people. The control group was offered water routinely in a day. In addition, the experimental group received 125 mL of dendrobe juice twice a day for four weeks. Quality of life for the old people was evaluated by the Patient Assessment of Constipation Quality of Life (PAC-QOL) Score and the symptom of constipation was assessed by Wexner score. Results: The four-week intervention brought significant performance improvement in all the measured parameters in the experimental group in comparison with the control group. These included significantly more frequency and shorter durations of defecation ($p < 0.01$, respectively), improved quality of life based on constipation score (PAC-QOL) (experimental group: $50.41 \pm 3.46$ vs. control group: $70.25 \pm 2.35$; $p < 0.05$), and improved score on the Wexner constipation scale (experimental group: $6.56 \pm 0.89$ vs. control group: $15.50 \pm 0.64$; $p < 0.05$). Conclusions: Fresh dendrobe beverage therapy is effective in improving stool frequency, reducing duration of defecation, and enhancing quality of life.

**Keywords:** fresh dendrobe juice; therapy; senile habitual constipation; older people; WeChat; social media

## 1. Introduction

Senile habitual constipation is one of the most frequent functional gastrointestinal disorders among older people, and the treatment of constipation causes significant impact on economic burden [1,2]. Senile habitual constipation is a common geriatric syndrome [3] that is caused by decline of body function, reduced strength of intestinal smooth muscle, and abdominal muscle [4]. The major symptom is difficulty with defecation, i.e., dry stools and reduced frequency of defecation.

With population aging and increasing social wealth, there is an increasing occurrence of senile habitual constipation in China. It is reported that approximately one third of people over 60 years old and about 80% of long-term bedridden older people suffer from constipation [5]. Long-term constipation not only negatively affects the health-related quality of life of older people, but also causes hemorrhoids, colorectal cancer, and other complications. Strain of defecation may even cause cardiovascular and cerebrovascular diseases and endanger the lives of the older people. Therefore, reducing constipation is an important component of nursing care for older people.

To date, there is no specific medicine to cure constipation. The common medical treatment is to use laxatives to facilitate bowel movement, which leads patients to drug

dependency. Diet therapy in traditional Chinese medicine appears to have a unique positive effect in prevention and treatment of constipation [6,7]. Dendrobe is known as "the first of the nine immortal herbs" in traditional Chinese medicine because it has high medical value (e.g., producing body fluids, nourishing the stomach or nourishing Yin) [8]. Modern pharmacological studies have also shown that dendrobe can protect and promote secretion of digestive juice and digestive enzymes such as saliva, gastric juice, and intestinal juice. It can also enhance gastrointestinal peristalsis, help digestion, and promote defecation [9]. Therefore, it has been recommended by the National Health Commission of China as food and drug for substance management and has been integrated into food in most provinces in China [10]. According to the traditional Chinese medicine literature, stems of dendrobe can be steamed and boiled to decoction for use to cure senile habitual constipation, but this method is often time consuming and complex. Direct chewing of fresh dendrobe can preserve most of the active ingredients, yet is difficult for older people to take with degrading swallow function [11]. Fresh dendrobe juice is easy to produce, can preserve the major medical ingredient, and is easy to swallow, and thus can be a feasible intake option [12].

Dendrobe is rich in mucus, which can promote secretion of gastric juice and help digestion [13]. The major useful ingredient of dendrobe juice is polysaccharides. Dendrobe polysaccharides can neither be decomposed by digestive enzymes nor generate energy, instead it can improve the activity of amylase and protease to optimize the intestinal peristalsis and environment, which further promote defecation, thus improving the symptoms of constipation [14,15]. However, there is a lack of empirical evidence to prove this theory. To address this knowledge gap, our study focuses on exploring the medical effect of the fresh dendrobe beverage therapy in curing habitual constipation for older people.

## 2. Materials and Methods

### 2.1. Research Design

Pilot research of a four-week trial (randomized and controlled) was carried out. A total of 56 participants were segregated into two groups at random. The experimental group received the fresh dendrobe beverage therapy. The control group was required to keep their normal daily activities and was guaranteed to receive the fresh dendrobe beverage therapy program after the end of the study. All parts of the research conform to the Declaration of Helsinki principles [16] and have the full ethical approval of the institutional ethics board of the Medical College, Hangzhou Normal University.

### 2.2. Sampling

2.2.1. Inclusion and Exclusion Criteria

Inclusion criteria for the study were: (1) ages between 60 and 70 years; (2) individuals diagnosed with senile habitual constipation using the Rome IV criteria [17,18]; (3) having a clear mind and communication abilities; (4) volunteering to take part in the research; (5) no history of taking laxatives or medical side effects, such as NSAID ibuprofen, Antidepressant Prozac, etc., and were constipated during the research period. The exclusion criteria were: (1) intestinal organic lesions; (2) illness restricting swallowing activity; (3) allergy to fresh dendrobe juice or suffering from mental disease (or neural or cardiovascular disease), which would jeopardize the informed decision making.

2.2.2. Sample Size Calculation

The effect size and statistical power of this study were calculated using SPSS–22 in reference to the study of Li and Liu [19], who conducted research in a similar population as ours. Based on this, the sample size of 28 older people per group was sufficient with the assumption that the attrition rate is 25%. The level of significance ($\alpha$) was set at 5%. Meanwhile, we set the test power ($1 - \beta$) at 80% for the research.

### 2.3. Participant Recruitment

Of this study, participants were recruited from Bai Taling Community in Hangzhou City, Zhejiang Province, China with the agreement of the community's management. The participants and their family members were informed about the study purpose and process when participants met the inclusion criteria. All participants were voluntary, and the written informed consent was acquired from older people or their family member before the study.

### 2.4. Randomization

In total, fifty-six older people met the inclusion criteria. They were divided into two groups (experimental group and control group, *n* = 28) randomly using a random number table.

### 2.5. Preparation of Fresh Dendrobe Juice

Fresh dendrobes were one-time selected and bought from a local market in Hangzhou. After washing with running water, the dendrobes were cut into 3 cm long pieces. Then, the dendrobes were put into the juicer. About 5 g of dendrobe was blended with 500 mL water. Dendrobe juice was produced by pressing the ingredients in a juicer (JYL-C91T, Hangzhou Joyoung Co., Ltd., Hangzhou, China).

### 2.6. Intervention

Participants in the experimental group and control group continued with their normal daily life, such as drinking water, eating, and exercising. In addition, 125 mL of fresh dendrobe juice was produced and delivered to the experimental group by the researcher every morning. They were asked to drink the dendrobe juice one hour after breakfast and lunch, twice a day for four weeks.

A social media group was created using the most popular Chinese social media platform WeChat for convenience of communicating with the participants and reminding them to drink the dendrobe juice twice a day, after breakfast and lunch. In China, the WeChat platform has been increasingly used by almost all people in all age groups. It has multiple functions for communication and is easy to use, i.e., text and voice messages, free voice and video calls, group chat, and Mini programs. All of the study participants had prior experience using WeChat to connect with friends and family and acquiring daily news on mobile phones or tablets.

Training was provided to all participants face-to-face by the researchers. The instructions included how to observe stool frequency, characteristics, and how to use the WeChat platform to record these data online every day. A researcher scanned a participant's personal QR code on WeChat to confirm their identities before providing dendrobe juice daily so as to ensure accurate identification of participants. In addition, a questionnaire was sent to each participant to fill and submit before 10 am each day via WeChat (As shown in Figure 1). The participants were encouraged to consult with the researchers by text and voice in the WeChat group when encountering any questions in consumption of the juice and filling in the questionnaire.

### 2.7. Measurement

The factors impacting defecation, i.e., dietary intake [20], dietary fiber [21], age, and gender [22], were collected upon enrolment from participants' personal health records one week before the intervention. Each score of participants on the Patient Assessment of Constipation Quality of Life questionnaire (PAC-QOL) and Wexner constipation were measured one week before the intervention and on a weekly basis after the intervention. The study period was 1–28 October 2020.

We determined defecation status of participants according to Rome IV Diagnostic Criteria before and after intervention. The stool form was graded using the Bristol Stool Form Scale (BSFS) [23] complied by Lewis and Heaton in 1997. This scale included seven

types of stool forms (1–7: 1 = dry and hard lumps and 7 = loose and watery) (as shown in Figure 2). The participants were taught to assess their stool form and consistency based on BSFS and fill in the questionnaire by clicking the relevant check box in the questionnaire mentioned above (As shown in Figure 3).

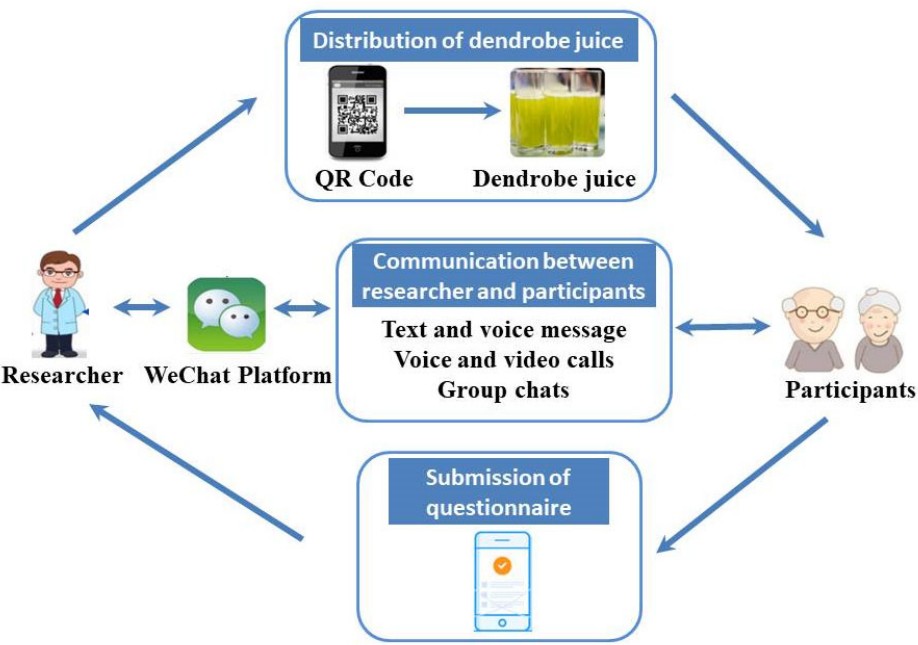

**Figure 1.** Distribution of the dendrobe juice and communication with the participating older people by the research team facilitated by the social media platform WeChat.

| Type 1 | | Separate hard lumps, like nuts (hard to pass) |
|---|---|---|
| Type 2 | | Sausage-shaped but lumpy |
| Type 3 | | Like a sausage but with cracks on its surface |
| Type 4 | | Like a sausages or a snake, smooth and soft |
| Type 5 | | Soft blobs with clear-cut edges (passed easily) |
| Type 6 | | Fluffy pieces with ragged edges, a mushy stool |
| Type 7 | | Watery, no solid pieces, entirely liquid |

**Figure 2.** The Bristol Stool Form Scale.

| Date | 2020.10.01 | | | | |
|---|---|---|---|---|---|
| Consumption of dendrobe juice | ☐0 | ☐1 | ☐2 | | |
| Defection frequency | ☐0 | ☐1 | ☐2 | ☐3 | ☐More than 3 |
| Character of stool | ☐ Type1–2 | ☐ Type3–4 | ☐ Type5–7 | | |
| Time taken to complete defecation | ☐ 0–10min | ☐ >10 min | | | |

**Figure 3.** The daily questionnaire about consumption of dendrobe juice and defecation.

### 2.8. Ability to Perform Quality of Life

The quality of life for older people was assessed by PAC-QOL questionnaire, and the severity of constipation was assessed by the Wexner constipation scale.

The validated PAC-QOL contains 28 questions grouped into 4 subscales. Questions 1–4 are about physical discomfort, questions 5–12 are about psychosocial discomfort, questions 13–23 are about worries and concerns, and questions 24–28 are about satisfaction. The data of the PAC-QOL questionnaire were recoded through a five-point Likert scale ranging from zero to four. A score of zero means 'not at all' or 'none of the time', while 4 represents 'extremely' or 'all of the time'. A lower score represented better quality of life [24]. The score on the Wexner constipation scale ranged from zero to thirty, with higher scores meaning severe symptoms [25].

### 2.9. Data Analysis

SPSS v22.0 packet (Chinese version, Chicago, IL, USA) was used for data analysis. The quantitative variables were expressed as mean $\pm$ standard deviation while the qualitative variables were described as frequency (number and percentage). For data that fits with normal distribution, the independent samples t test was used to compare means of independent variables, and chi-square test was used to compare categorical variables. $p < 0.05$ indicated a significant level in all tests.

## 3. Results

### 3.1. Participant Characteristics

Fifty-six eligible old people with constipation were enrolled in the study: 23 males and 33 females.

In the experimental group, one participant withdrew from the study citing inconvenience due to a changing domestic arrangement at the end of the four weeks. Finally, 27 older people constituted the experimental group (12 males and 15 females), and 28 older people constituted the control group (10 males and 18 females), both completed the four-week trial. There were no significant differences in age ($p = 0.77$), gender ($p = 0.59$), educational level ($p = 0.56$) or other basic information between the two groups. The ages of the participants ranged from 60 to 70 years with an average age of 64.78 years. In China, retirement age is 60 years, and people over 60 years of age are seen as older people. The participants reported their questionnaires through the WeChat Mini program every day. Then questionnaires submitted could also be checked through the Mini program in WeChat by the researchers; there was no instances of under-reporting or over-reporting or on-reporting. No significant extreme values and contradictory data were found in the data collected.

Both groups have a similar level of education: 70.37% in the experimental group have a junior high school education (10-year education), more than that of the 57.14% in the control group. Participants in the experimental group spent an average of 14.2± 1.6) minutes per time defecating to complete defecation and a total number of (1.4 ± 0.5) times per week defecating one week before the intervention. The control group spent an average of (13.46 ± 1.40) minutes per time defecating to complete defecation and a total number of (1.43 ± 0.50) times per week defecating. There were no significant inter-group differences at baseline for any parameter (As shown in Table 1).

**Table 1.** Baseline characteristics of research participants.

| Variables | Experimental Group (*n* = 27) | Control Group (*n* = 28) | $t/X^2$ | $p$ |
|---|---|---|---|---|
| Age (years) | 64.89 ± 2.10 | 64.68 ± 3.13 | −0.29 * | 0.77 |
| Gender, *n* (%) | | | | |
| Male | 12 (44.44) | 10 (35.71) | 0.437 † | 0.59 |
| Female | 15 (55.56) | 18 (64.29) | | |
| Education, *n* (%) | | | | |
| primary school | 6 (22.22) | 8 (28.57) | | |
| Junior high | 19 (70.37) | 16 (57.14) | 1.20 † | 0.56 |
| ≥Senior high | 2 (7.41) | 4 (14.29) | | |
| Total No. of defecation per week | 1.44 ± 0.51 | 1.43 ± 0.50 | −0.12 * | 0.91 |
| Average completion time (min) | 14.19 ± 1.59 | 13.46 ± 1.40 | −1.78 * | 0.08 |
| Stool form, *n* (%) | | | | |
| Dry and hard (Type 1–2) | 24 (61.54) | 26 (65.00) | | |
| Normal (Type 3–4) | 15 (38.46) | 14 (35.00) | 0.10 † | 0.82 |
| Loose (Type 5–7) | — | — | | |

Notes: * Student's *t*-test. † Chi-square test.

### 3.2. Frequency, Duration, and Characteristics of Defecation

Figure 4 illustrates the change in the average number of defecations per person per week in both groups over the four-week experimental period. There were no statistically significant differences between the two groups in the frequency of defecation at the start point of the experiment (T0, from 24 to 30 September). The frequency of defecation increased steadily in the experimental group. A significant group difference appeared in the third week of the experiment (from 15 to 21 October, $p < 0.05$). The frequency of defecation in the experiment group in the fourth week (from 22 to 28 October) was substantially higher than that in the first week ($p < 0.01$). Meanwhile, frequency of defecation did not change significantly in the control group over the four-week period.

For the experimental group, there were, on average, 11 defecation times recorded in the four-week experiment period, that is, 0.39 times per day; both were significantly much more preferable ($p < 0.01$) than those of the control group, which had on average 5.8 defecation times in the four-week experiment period, or 0.21 times per day. The average time taken to complete a defecation in the experimental group was less than that taken by the control group (10.97 ± 3.66 vs. 13.05 ± 2.13, $p < 0.01$). For most participants (73.31%) in the experimental group, their stool type was normal, but stool type of more than half of the participants (55.83%) were dry and hard in the control group ($p < 0.01$) (Table 2).

### 3.3. Comparison of the PAC-QOL Score between the Two Groups and over the Time of the Intervention

The PAC-QOL score of the experimental group was (72.37 ± 2.94), similar with that of the control group (71.57 ± 2.81) at the start of the experiment. The total score of PAC-QOL was (50.41 ± 3.46) in the experimental group at the end of week 4, which was significantly lower than that of the control group (70.25 ± 2.35, $p < 0.01$), indicating an improvement in quality of life (As shown in Figure 5 and Table 3).

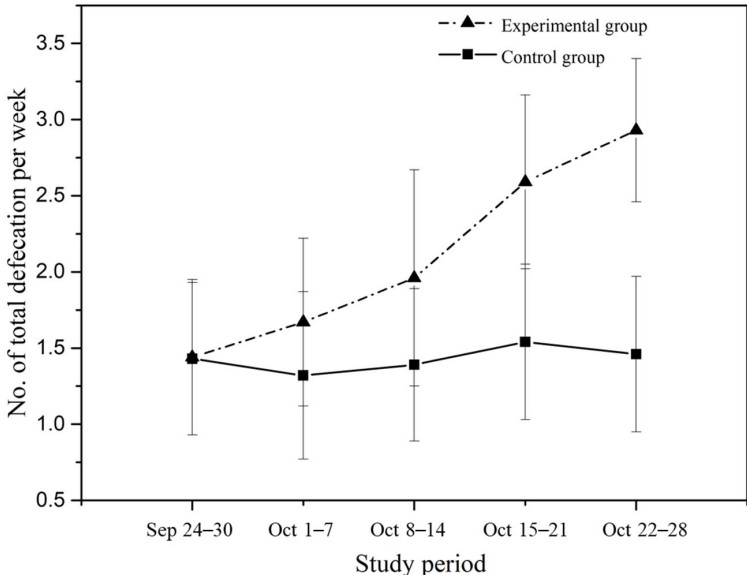

**Figure 4.** Comparison of the weekly change in total defecation times per person between the two groups in the four-week study period.

**Table 2.** Comparison between measurement items between the Experiment Group and the Control Group.

| Measured Item for Each Person | Experimental Group (*n* = 27) | Control Group (*n* = 28) | Test Statistics | *p* |
|---|---|---|---|---|
| Total No. of defecation in four weeks | 11.00 ± 2.86 | 5.80 ± 1.83 | −5.06 * | <0.001 |
| No. of defecation per day | 0.39 ± 0.10 | 0.21 ± 0.07 | −4.89 * | <0.001 |
| Time taken to complete defecation, *n* (%) | | | | |
| 0–10 min | 161 (54.39) | 19 (11.66) | | |
| >10 min | 135 (45.61) | 144 (88.34) | 34.89 † | <0.001 |
| Average completion time(time/minutes) | 10.97 ± 3.66 | 13.05 ± 2.13 | | |
| Stool form, *n* (%) | | | | |
| Dry and hard (Type 1–2) | 51 (17.23) | 91 (55.83) | | |
| Normal (Type 3–4) | 217 (73.31) | 72 (44.17) | 80.22 † | <0.001 |
| Loose (Type 5–7) | 28 (9.46) | — | | |

Notes: * Student's *t*-test. † Chi-square test.

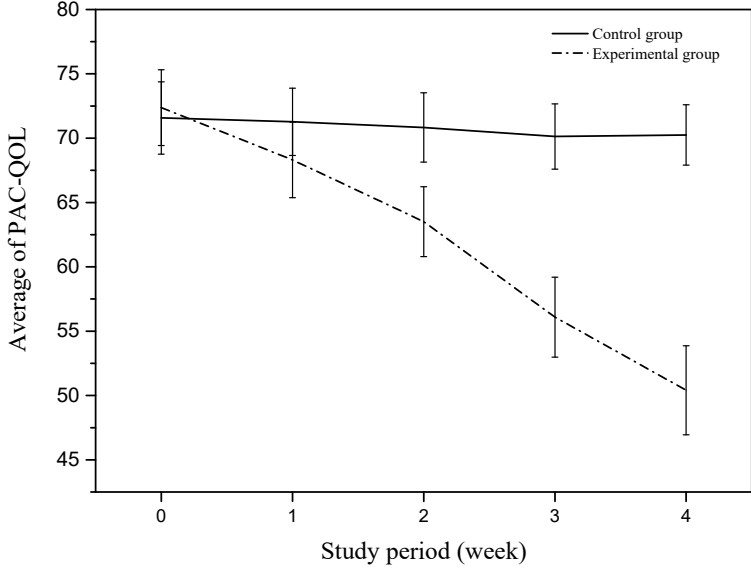

**Figure 5.** Score on scale of PAC-QOL of constipation patients in the four-week study period.

**Table 3.** Comparison of the PAC-QOL score between the intervention and the control group before and after the intervention (case, $\overline{x} \pm$ S).

| Domain | Before Intervention | | After Intervention | |
|---|---|---|---|---|
| | Experimental Group | Control Group | Experimental Group | Control Group |
| Body discomfort | 11.74 ± 1.23 | 11.25 ± 1.17 | 8.22 ± 1.34 ** | 10.92 ± 1.05 |
| Psychosocial discomfort | 20.48 ± 2.14 | 19.96 ± 1.90 | 16.04 ± 1.26 ** | 19.75 ± 1.67 |
| Worry and anxiety | 25.96 ± 1.37 | 26.00 ± 1.36 | 17.67 ± 2.70 ** | 25.50 ± 1.20 |
| Satisfaction | 14.19 ± 1.62 | 14.36 ± 1.37 | 8.48 ± 2.03 ** | 14.07 ± 1.30 |
| Total scores | 72.37 ± 2.94 | 71.57 ± 2.81 | 50.41 ± 3.46 ** | 70.25 ± 2.35 |

Notes: The experimental group and control group were compared before the intervention. The control group was compared before and after the period of the four-week experiment. The experimental group and control group were compared after the intervention, ** $p < 0.01$. The experimental group was compared before and after the period of the four-week experiment, ** $p < 0.01$.

### 3.4. Score on Wexner Constipation Scale before and after Intervention

The Wexner constipation score was similar between the two groups at the start of the experiment (As shown in Table 4). It was significantly lower in the experiment group than that of the control group (6.56 ± 0.89 vs. 15.50 ± 0.64, $p < 0.05$) after the intervention. The Wexner constipation score of the experimental group also showed significant difference before and after the experiment (15.70 ± 1.14 vs. 6.56 ± 0.89, $p < 0.05$), whereas that in the control group remained similar (As shown in Figure 6).

**Table 4.** Wexner constipation score before and after intervention in two groups.

| Group | n | Before Intervention | After Intervention | Difference (95% CI) | t | p |
|---|---|---|---|---|---|---|
| Experimental group | 27 | 15.70 ± 1.14 | 6.56 ± 0.89 | 8.81 (7.93–9.69) | 20.01 | <0.001 |
| Control group | 28 | 15.54 ± 1.07 | 15.50 ± 0.64 | 0.02 (−0.47–0.51) | 0.08 | 0.94 |
| Difference (95% CI) | | −0.17 (−0.77–0.43) | 8.94 (8.53–9.36) | | | |
| t | | −0.56 | 42.90 | | | |
| P | | 0.58 | <0.001 | | | |

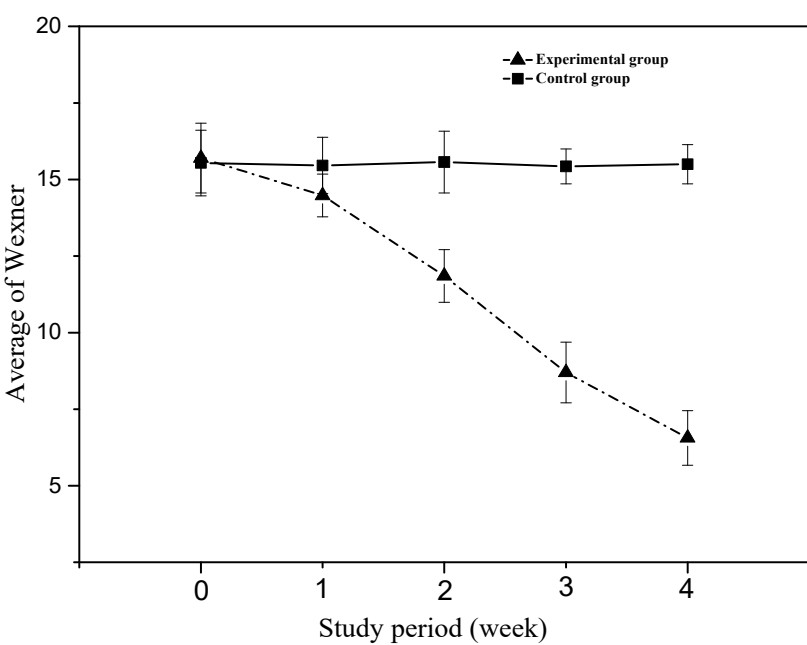

**Figure 6.** Score on the Wexner constipation scale in the four-week study period.

## 4. Discussion

Using the method of randomized controlled trial, this study designed, implemented, and tested the effect of fresh dendrobe beverage therapy on relieving constipation for older people. It measured the outcome in terms of PAC-QOL and Wexner score.

Traditional Chinese medicine believes that constipation is caused by weakness of the spleen and stomach in the human body [26]. Dendrobe has the function of clearing away heat and can relieve failing of the spleen and stomach. Its major advantage is that it is without significant side effect during the period of consumption [19,27]. This corresponded with our case. There were no side effects reported among all the participants during the experimental period. The experiment results suggest that consumption of fresh dendrobe juice can bring improvement for individuals with senile habitual constipation. This result is consistent with Li's study [19]. They administered fresh dendrobe to 41 older people with senile habitual constipation in a hospital for one month. The intervention included participants who ate 5 g fresh dendrobe per day. They found 65% of the older people experienced improvement in intestinal function and softened stool. In our study, the average defecation time in the experimental group decreased from $14.19 \pm 1.59$ to $10.97 \pm 3.66$, and the PAC-QOL scores declined from $72.37 \pm 2.94$ to $50.41 \pm 3.46$. This provides evidence to demonstrate the improvement of life in the experimental group after intervention. The difference of our design from Li was pressing fresh dendrobe and administering it as a beverage. With many fibers, directly eating dendrobe [19] is difficult for older people with poor chewing and digestive functions. Therefore, we processed fresh dendrobe into juice, which is safer and convenient to administer and easier for older people to accept. Our study also validates the viewpoint of professor Mao [28], who suggested that fresh dendrobe has the function of replenishing vital essence, nourishing the intestinal system, and improving gastrointestinal functions, which can lead to relieving constipation. We believe this is related to the active constituent of dendrobe. The main components of dendrobe are polysaccharides, which have the function of protecting the gastrointestinal tract and enhancing gut microbiota stability [29]. Some literature [14,30] reports that dendrobe contains organic acids. These organic acids can reduce pH in the intestinal environment, stimulate intestinal peristalsis, and improve constipation symptoms. As shown in Table 2, the defecation time of the experimental group steadily decreased in the four-week experiment period, and the total number of defecations per week was much more than that of the control group. These results suggest that fresh dendrobe juice can relieve constipation symptoms for older people with difficulty of defecation. We believe this is related to the Wexner constipation score, PAC-QOL score, and defecation frequency per person, which were all significantly improved in the experimental group after the intervention. The PAC-QOL questionnaire is a valid and reliable tool for quality-of-life assessment in participants with chronic constipation and the Wexner constipation score is a uniform tool for assessment of chronic functional constipation [31]. Therefore, these changes suggest a reduction in the bloated, uncomfortable, and sick symptoms of the participants with constipation.

By June 2020, China had 940 million Internet users, of whom 10.3% are over 60 years old [32]. The WeChat platform has become a popular social media platform in China. It has been used widely by Internet users in their daily life with the increasing penetration of smart phones in China since 2011 [33]. Previous studies have found that WeChat and other modern network tools have been used to improve the level of nursing management and overall nursing quality [34]. In this study, we used the WeChat platform to replace paper records for older people to self-record and report the outcome data after drinking dendrobe juice on a daily basis. We used the real-time video and voice communication channel to provide orientation and guidance to the study participants on the data recording and uploading method and procedure and responded to their questions through WeChat phone call and messaging on a timely basis. These social media interactions reduced the complexity and cost of conducting this experiment in the community, where older people could participate in the dietary therapy in their comfortable home and self-capture and

report outcome data through the WeChat social media. It also ensured accuracy of reporting without missing data. This high-quality data collection suggests that using WeChat for communication and self-collection of data by older people is a feasible method for health interventions targeting community-dwelling older people with minimum cost and effort. In future studies, we can further improve the program interface and content design to improve understandability and facilitate its use for older people to use it more easily and effectively.

## 5. Study Limitations

The first limitation was that this case study was conducted in one community in Hangzhou city, China, as it was a pilot study, limiting the generalizability of the findings to different population groups. Secondly, according to previous research, most studies about Chinese medicine's influence on constipation were conducted within one month [35,36]. However, our intervention period was four weeks, which may still not be long enough for better effects to be achieved. The experiment was conducted in autumn, when constipation rates were high in the study region [37]. Therefore, the findings may be influenced by season. To ensure that the study findings are generalizable to other seasons, further research can be conducted in different seasons.

## 6. Conclusions

This randomized controlled trial found that fresh dendrobe beverage therapy administered twice per day can significantly improve symptoms of senile habitual constipation and overall PAC-QOL for people over 60 years of age. It provides evidence to suggest that dendrobe juice can be considered for inclusion in the diet plan for older people with constipation. Future study can focus on prolonging the time of consumption of fresh dendrobe juice to achieve the aim of a radical cure for senile habitual constipation. The study also demonstrates the effectiveness of study participants using the WeChat platform to capture and report experimental outcome data on a daily basis, which is an innovative approach with great potential to reduce complexity and cost of data collection for consumer-facing health interventions.

**Author Contributions:** Writing—original draft preparation, Y.L.; writing—review and editing, S.C.; investigation, M.W.; resources, D.H.; data curation, Y.C. and B.Q. All authors have read and agreed to the published version of the manuscript.

**Funding:** This research was supported by the First-class Discipline Project of Zhejiang Province (No. 4065C4011700201), China; First-class Discipline Project of Qianjiang College of Hangzhou Normal University (No. 2019JXYL003), China; the Policy Theory Research Project of Zhejiang Provincial Civil Affairs Department (No. ZMKT202140), China; Medical and Health Technology Plan of Zhejiang Province (No. 2022507615); Department of Higher Education, Ministry of Education, 2021 Industry University Research Cooperation and Education Project (202102627005); Research Project of Zhejiang Higher Education Association (KT2021191).

**Institutional Review Board Statement:** Approval was obtained from the ethics committee of Hangzhou Normal University. The procedures used in this study adhere to the tenets of the Declaration of Helsinki.

**Informed Consent Statement:** Informed consent was obtained from all subjects involved in the study.

**Data Availability Statement:** Data is contained within the article.

**Conflicts of Interest:** The authors declare no conflict of interest.

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
