# Peer review of "Effect of Fresh Dendrobe Juice Consumption on Senile Habitual Constipation for Older People: A Four-Week Randomized Controlled Trial"

_sustainability, doi:10.3390/su14063656_

Round 1

Reviewer 1 Report

Page 2. 2.2.1 Inclusion and exclusion criteria

  1. One of the participants inclusion criteria was aged over 60. What age range did the participants spam?
  2. The aged participants might suffer chronic diseases and polypharmacy. Some of the medicine side effects, such as NSAID ibuprofen, Antidepressant Prozac, etc. were constipation. Did authors take these factors into consideration?

Page 3. 2.5 Preparation of fresh dendrobe juice

  1. Did authors verify the origin of the dendrobe for the batch consistency in this study?
  2. The dendrobe was cut into 3 cm long pieces. It might be difficult for swallowing in aged participants. How is the adherence during the study? Did the adherence affect the study results?

Page 10. Study limitation and Conclusion

The constipation happens due to complex reasons, such as inadequate intake of fiber, changes in diet or routine and bowl movement. It is also gender dependent, especially in pregnant and after childbirth. Only 4 weeks intervention might be not long enough to improve the participants’ constipation. Can authors provide more solid evidence to support the conclusion.

Reviewer 2 Report

The study provides an advance in the field of habitual constipation treatment. Moreover, the results are important considering the life quality improvment and risks associated with chronic constipation in older people.

The manuscript is easy to follow, the methods are clear, results match the methods, and figures and tables are clear, and the conclusions correlate to the results.

Minor editing corrections are needed.

Reviewer 3 Report

  1. The abbreviation of PAC-QOL should be explained when it is firstly used in text, line 139
  2. Trial was conducted from October 1, 2020 to October 28, 2020, authors consider Rome III criteria, but since May 2016 there are Rome IV criteria. I recommend to use the latest one.
  3. Please provide citation for sentences for line 258 and 259
  4. There is no connection between references and text in lines 261-276
  5. Discussion is mostly focused on advantages of using WeChat in this research. However I recommend to discuss more thoroughly dendrobium juice in medicine. To diverse references I suggest citing most recent Western countries literature, such as

Włodarczyk, J.; Waśniewska, A.; Fichna, J.; Dziki, A.; Dziki, Ł.; Włodarczyk, M. Current Overview on Clinical Management of Chronic Constipation. J. Clin. Med. 2021, 10, 1738. https://doi.org/10.3390/jcm10081738

Mari A, Mahamid M, Amara H, Baker FA, Yaccob A. Chronic Constipation in the Elderly Patient: Updates in Evaluation and Management. Korean J Fam Med. 2020 May;41(3):139-145. doi: 10.4082/kjfm.18.0182. Epub 2020 Feb 17. PMID: 32062960; PMCID: PMC7272371.

Yang L, Wan Y, Li W, Liu C, Li HF, Dong Z, Zhu K, Jiang S, Shang E, Qian D, Duan J. Targeting intestinal flora and its metabolism to explore the laxative effects of rhubarb. Appl Microbiol Biotechnol. 2022 Feb;106(4):1615-1631. doi: 10.1007/s00253-022-11813-5. Epub 2022 Feb 7. PMID: 35129656.

  1. There is no information about side effects of dendrobium juice.
  2. It should be mentioned that it is pilot research due to small number of participants.
  3. All references should be checked once again because most of them are not connected with the text and the doi address is not correct.

Round 2

Reviewer 1 Report

The authors' response is acceptable. This article would be suitable for publication.

Reviewer 3 Report

Authors applied changes with accordance to suggestions. I recommend to accept this article.